# LoRA is All You Need for Safety Alignment of Reasoning LLMs

## Abstract

Reasoning LLMs have demonstrated remarkable breakthroughs in solving complex problems that were previously out of reach. To ensure LLMs do not assist with harmful requests, safety alignment fine-tuning is necessary in the post-training phase. However, safety alignment fine-tuning has recently been shown to significantly degrade reasoning abilities, a phenomenon known as the "Safety Tax". In this work, we show that using LoRA for SFT on refusal datasets effectively aligns the model for safety without harming its reasoning capabilities. This is because restricting the safety weight updates to a low-rank space minimizes the interference with the reasoning weights. Our extensive experiments across four benchmarks covering math, science, and coding show that this approach produces highly safe LLMs—with safety levels comparable to full-model fine-tuning—without compromising their reasoning abilities. Our ablation studies further identify three key factors in LoRA: (1) rank-1 updates are sufficient to achieve the best reasoning and safety performance, (2) the up projection layers are the most critical modules, with LoRA applied to them alone achieving even better results, and (3) middle layers are more effective than early or late layers. Together, these findings show that strong safety and reasoning can be achieved at minimal computational cost when updates are applied in the right places. Additionally, we observe that LoRA induces weight updates with smaller overlap with the initial weights compared to full-model fine-tuning. Finally, while our attempts to further reduce this overlap yield only modest improvements on some tasks, they highlight the potential of developing methods that more reliably optimize the reasoning–safety tradeoff.

## 1 Introduction

Large language models (LLMs) have made remarkable progress across a wide range of tasks. A major recent breakthrough is the emergence of LLMs with advanced reasoning capabilities, enabling them to solve complex problems previously out of reach. However, recent studies have reported significant safety risks associated with reasoning-capable models (Jiang et al., 2025; Zhou et al., 2025; Huang et al., 2025; Li et al., 2025a). Indeed, reasoning fine-tuning—the process through which LLMs acquire these capabilities—often compromises safety, even when starting from a safety-aligned checkpoint (Jiang et al., 2025; Zhou et al., 2025; Zhao et al., 2025; Li et al., 2025a). For example, Jiang et al. (2025) show that models distilled for reasoning from DeepSeek-R1 become substantially less safe than their original base models.

There has been significant effort in the literature to preserve LLMs' safety alignment during instruction fine-tuning. However, these approaches are not applicable to reasoning fine-tuning. First, reasoning fine-tuning datasets are often highly curated (Muennighoff et al., 2025) and unlikely to contain unsafe content. Thus, data filtering methods such as those proposed by Shen et al. (2024); Choi et al. (2024); Bianchi et al. (2023) are not applicable. In addition, methods that restrict model updates during fine-tuning (Hsu et al., 2024; Mukhoti et al., 2023) are ineffective in the reasoning setting, as acquiring reasoning capabilities typically requires longer training and more substantial weight updates compared to instruction fine-tuning. To the best of our knowledge, the current literature does not offer any method for safety alignment of reasoning models.

Instead, the prevailing strategy is to apply a secondary safety alignment phase after reasoning capabilities have been acquired. This phase—often implemented via supervised fine-tuning (SFT) or reinforcement learning (RL)—has become a standard step in modern LLM development. Although safety alignment fine-tuning can significantly improve model safety, it often comes at a steep cost to reasoning performance—a phenomenon referred to as the "Safety Tax" (Huang et al., 2025). Even incorporating chain-of-thought (CoT) style reasoning into safety fine-tuning datasets (Jiang et al., 2025) cannot succeed in fully preserving reasoning abilities (Huang et al., 2025).

In this work, we investigate the algorithmic factors that contribute to this trade-off. Existing evidence suggests that safety-related behavior in LLMs is often governed by a small number of dominant directions—either in activation space, such as steering vectors (Panickssery et al., 2023) or refusal features (Arditi et al., 2024; Yu et al., 2024), or in weight space. In particular, Jain et al. (2024); Wei et al. (2024) show that safety-critical weights tend to lie in a low-rank subspace. In our analysis, we find that the model undergoes relatively high-rank changes during full-model fine-tuning (see Figure 1), which results in Safety Tax. This highlights a key insight: although achieving safety may require modifying weights only along a low-rank subspace, full-model fine-tuning permits arbitrary updates, potentially introducing many unnecessary changes that interfere with reasoning.

Our extensive experiments reveal the surprising effectiveness of a simple recipe for safety alignment of reasoning models: Applying LoRA during SFT using a straightforward direct refusal data set. Despite its simplicity, this approach achieves safety performance on par with full-model alignment, while preserving reasoning capability close to that of the original reasoning-tuned model. This result holds for both 7B and 14B models and is validated across four benchmarks spanning mathematics, science, and code generation. It represents a rare "one stone, three birds" outcome: strong safety, strong reasoning, and computational efficiency.

Moreover, we further ablate the LoRA configuration to understand "how much LoRA is sufficient". We make three key findings. (1) Setting the rank to $r = 1$ achieves the best reasoning–safety tradeoff (in terms of the Pareto frontier when the number of training epochs is varied). This is encouraging, as it shows that strong performance on both reasoning and safety can be achieved at the lowest possible fine-tuning cost. (2) Updating only the up projection layers in the MLP yields an even better tradeoff than updating the full MLP, while updating only the gate or down projections performs worse. This highlights the central role of the up projection and motivates future research into understanding why it is so effective. (3) Middle layers are more important for a good reasoning–safety tradeoff: updating only 16 middle layers is often sufficient, whereas updating early or late layers yields worse results. Interestingly, this connects to prior findings that safety-critical features often emerge in the middle layers of LLMs (Panickssery et al., 2023; Arditi et al., 2024). Overall, our results provide meaningful insights into the key elements of LoRA configuration that affect the reasoning–safety tradeoff, and can help achieve strong performance at minimal cost.

Additionally, we explore the weight structure imposed by LoRA to understand the differences it introduces. We find that LoRA updates are not only low-rank by design but also exhibit smaller alignment with the weights of the original reasoning model compared to those from full-model fine-tuning—across most layers. While the reduction in alignment is small, it may suggest that LoRA updates are less disruptive to reasoning-related weights. We further investigate whether explicitly reducing such overlap—via regularization or post-hoc weight merging—can improve safety or reasoning capabilities. We find that one post-hoc method for reducing overlap achieves a modest gain in the reasoning–safety trade-off on some tasks. This shows promise, but more effort is needed to develop approaches that yield consistent improvements across tasks, which we consider a valuable direction for future work.

## 2 RELATED WORK

To develop LLMs that are both safe and capable, models can be safety aligned before or after fine-tuning.

**Fine-tuning a safety-aligned model.** Fine-tuning a safety-aligned model often leads to safety degradation. For instruction fine-tuning, safety degradation is shown across various model architectures and optimization strategies, including full-model and LoRA fine-tuning (Qi et al., 2023; Hsiung et al., 2025; Zhan et al., 2023). To mitigate this issue, several methods have been proposed.

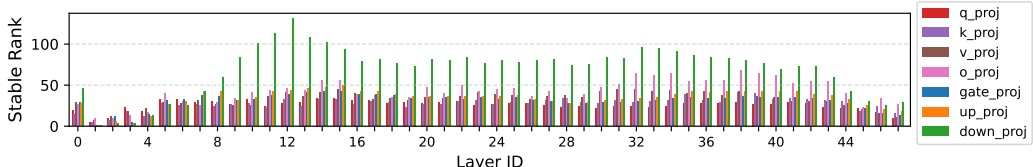

Figure 1: We compute the stable rank of the difference between the full-model fine-tuned model's weights and those of the original `DeepSeek-R1-Distill-Qwen-14B` for each layer. Here, the colors indicate the module types, and the x-axis shows the layer indices. We observe that the stable rank is quite high—ranging from around 40 to 100 for most layers

Shen et al. (2024); Choi et al. (2024) focus on data filtering, aiming to remove unsafe examples from the fine-tuning data. Bianchi et al. (2023) shows that injecting just a few hundred safety examples during instruction fine-tuning can improve safety. Peng et al. (2025) leverages an existing guardrail model to encourage safe response segments while suppressing unsafe ones. Lyu et al. (2024) emphasizes the importance of prompt templates in preserving safety. Algorithmic approaches have also been explored: Hsu et al. (2024) propose projecting LoRA updates into a "safety subspace" derived from differences between aligned and unaligned models, while Mukhoti et al. (2023) introduce regularization techniques to constrain changes in intermediate representations during fine-tuning.

This approach is, however, not applicable to reasoning models. This is because acquiring reasoning capabilities typically requires longer training and more substantial weight updates compared to instruction fine-tuning. This results in losing the initial safety alignment of the model. Indeed, several recent studies have reported significant safety risks associated with reasoning-capable models Jiang et al. (2025); Zhou et al. (2025); Huang et al. (2025); Li et al. (2025a).

**Safety alignment after fine-tuning.** Aligning a fine-tuned model is typically done using supervised fine-tuning (SFT) and/or reinforcement learning (RL) (Wei et al., 2021; Griffith et al., 2013; Dai et al., 2023; Ouyang et al., 2022; Rafailov et al., 2023; Bai et al., 2022; Guan et al., 2024). However, this approach introduces a key trade-off: safety alignment can substantially impair model performance. Huang et al. (2025) characterizes this as the "Safety Tax," showing that safety-aligned models often lose much of their reasoning ability. Jiang et al. (2025) attempted to address this issue by constructing a safety fine-tuning dataset with long chain-of-thought (CoT) responses, but the resulting models still showed noticeable drops in reasoning performance (Huang et al., 2025).

In our work, we show that a simple application of LoRA for safety alignment of reasoning models can effectively align reasoning-capable models without compromising their performance.

## 3 LoRA for Safety Alignment Without Compromising Reasoning

In this section, we investigate whether the "Safety Tax" (Huang et al., 2025) can be mitigated. In particular, we aim to answer the following question: can we align a model for safety without compromising its reasoning capabilities? We focus on the same setup as in (Huang et al., 2025), where SFT is performed on safety datasets that provide harmful requests paired with refusal responses.

Our key observation is that during full-model fine-tuning, which is used in (Huang et al., 2025), the weights undergo relatively high-rank changes. As shown in Figure 1, we observe high stable ranks for the weight updates, i.e., differences between the fine-tuned model's weights and those of the initial model, in most layers. However, prior evidence suggests that safety behavior in LLMs is typically governed by only a single or a few directions in the activations (Panickssery et al., 2023; Arditi et al., 2024) and weights (Wei et al., 2024; Jain et al., 2024), indicating that a small low-rank modification may be sufficient to induce safe behavior, without altering the entire weight space. Thus, we conjecture that the degradation in reasoning performance is caused by full-model fine-tuning introducing unnecessary changes in many directions, which interfere with critical weights responsible for reasoning.

To address this, we consider Low-Rank Adaptation (LoRA) (Hu et al., 2022), originally proposed as a parameter-efficient fine-tuning method to reduce training cost and memory usage. Rather than

updating the full weight matrices, LoRA injects trainable low-rank matrices into existing layers while keeping the original weights frozen. Formally, a weight matrix $\boldsymbol{W} \in \mathbb{R}^{d \times k}$ is modified as:

$$\boldsymbol{W}' = \boldsymbol{W} + \Delta \boldsymbol{W}, \quad \text{where} \quad \Delta \boldsymbol{W} = \frac{\alpha}{r} \boldsymbol{B} \boldsymbol{A}, \tag{1}$$

where $\boldsymbol{B} \in \mathbb{R}^{d \times r}$ and $\boldsymbol{A} \in \mathbb{R}^{r \times k}$ are the trainable low-rank matrices with $r \ll \min(d, k)$, and $\frac{\alpha}{r}$ is the scaling factor, with $\alpha$ being a hyperparameter.

LoRA is particularly well-suited to our needs: it restricts updates to a low-rank subspace, thereby significantly reducing interference with the original weights. We will show in our experiments that this method works excellently, enabling the model to become safe while maintaining strong reasoning performance across benchmarks. As an additional benefit, LoRA is significantly more computationally efficient than full-model fine-tuning.

## 4 LoRA Bypasses the "Safety Tax"

In this section, we first introduce our safety fine-tuning and evaluation pipeline. Then, we evaluate models' safety alignment and reasoning performance after full-model and LoRA safety fine-tuning.

### 4.1 Safety Alignment Fine-Tuning of Reasoning LLMs

We begin with a reasoning-capable language model and perform safety alignment fine-tuning. Specifically, we apply supervised fine-tuning (SFT) on a safety dataset consisting of harmful questions paired with refusal responses, aiming to teach the model to reject harmful requests. We choose SFT over reinforcement learning (RL)-based techniques because it is simpler, less expensive, and does not require additional components such as a reward model. However, we expect our results to apply to RL safety alignment as well.

In our training setup, we compare two approaches: (1) **full-model fine-tuning**, as in (Huang et al., 2025), where all model parameters are updated using standard gradient-based optimization; and (2) **LoRA fine-tuning**, as described in Section 3.

### 4.2 Evaluation of the Fine-Tuned Model

After safety alignment fine-tuning is completed, we evaluate two aspects of the model: (1) safety, which is assessed using a dataset of harmful questions. We sample responses from the model for these questions and use `Llama-Guard-3-8B`—a model specialized in safety evaluation and shown to be the strongest safety evaluator in (Jiang et al., 2025)—to determine whether the responses are safe. The *safety score* is defined as the proportion of questions for which the model's response is judged to be harmful. (2) reasoning ability, evaluated using multiple standard benchmark datasets containing questions on math, science, and coding—widely used to assess models' reasoning capabilities. We consider the commonly used metric Pass@1 to measure accuracy on these benchmarks. For each question, we sample $n$ responses, compute the fraction of correct responses, and then average this accuracy over all questions. We set $n = 8$.

### 4.3 Datasets and Models

**Models.** We conduct experiments on two widely used open-weight reasoning-capable models: `DeepSeek-R1-Distill-Qwen-7B` and `DeepSeek-R1-Distill-Qwen-14B`. Safety evaluation is performed using `Llama-Guard-3-8B`, which was found to be the most accurate evaluator in (Jiang et al., 2025).

**Safety fine-tuning dataset.** We use the `DirectRefusal` dataset, adapted from (Rosati et al., 2024) by (Huang et al., 2025), which provides harmful requests paired with refusal-style answers.

**Safety evaluation dataset.** We adopt the `StrongREJECT` dataset (Souly et al., 2024), which consists of 310 policy-violating queries designed to test whether the model behaves safely.

**Reasoning benchmarks.** We evaluate the models performance on (1) American Invitational Mathematics Examination 2024 (`AIME`), (2) `GPQA` (Rein et al., 2024) evaluate mathematical and scientific

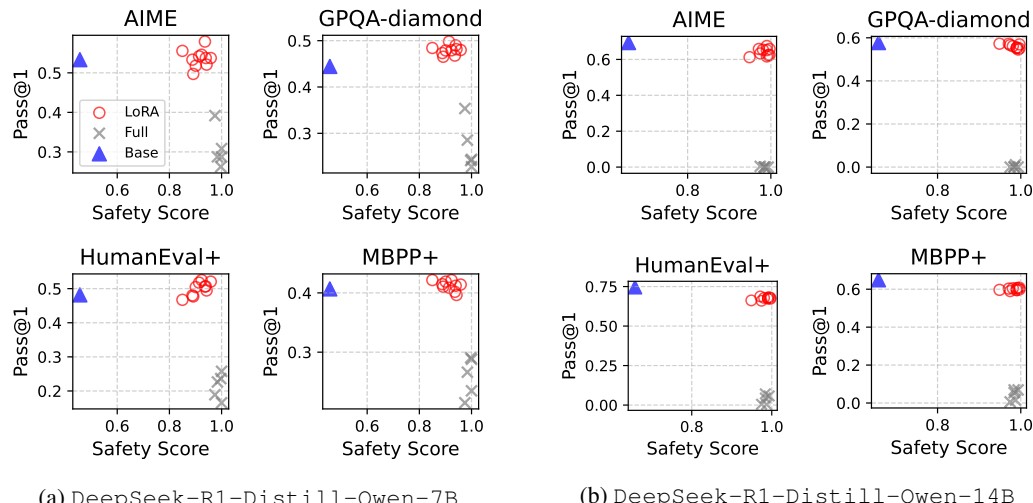

(a) `DeepSeek-R1-Distill-Qwen-7B`  (b) `DeepSeek-R1-Distill-Qwen-14B`

Figure 2: LoRA bypasses the "Safety Tax", achieving safety comparable to that of the full-model fine-tuned model and reasoning performance comparable to the original reasoning model. We plot reasoning performance—measured by Pass@1—against safety scores for different models. For the fine-tuned models, we report results for checkpoints at all epochs. Results on the base versions of `HumanEval` and `MBPP` are provided in Figure 9 in the Appendix, where the same patterns hold, but with higher accuracy.

reasoning, respectively, (3) `HumanEval` (Chen et al., 2021) and (4) `MBPP` (Austin et al., 2021) are code generation benchmarks. We also consider the augmented versions created by `EvalPlus` (Liu et al., 2023), denoted as `HumanEval+` and `MBPP+`.

**Training Setup.** Full-model fine-tuning is performed for 5 epochs, while LoRA fine-tuning is run for 10 epochs. We save and evaluate checkpoints at every epoch. Unless otherwise stated, LoRA is applied only to the MLP layers with rank $r = 1$. In Section 5, we investigate the effect of varying $r$ and applying LoRA to different modules (e.g., MLP vs. attention) and layers.

Additional experimental details are deferred to Appendix A.

### 4.4 LoRA is All You Need for Safety Alignment of Reasoning LLMs

Figure 2 compares the safety and reasoning capabilities at different checkpoints (i.e., epochs), during full-model and LoRA safety alignment fine-tuning. We observe that the base model before safety fine-tuning exhibits high accuracy but low safety. On the other hand, the fully fine-tuned models achieve good safety at the cost of reduced accuracy. In contrast, the LoRA fine-tuned models maintain strong performance in both safety and reasoning (as evidenced by the red points in the upper-right corner of the plot).

For the 7B models, the best LoRA checkpoints outperform the base model on all tasks. In terms of safety, they fall slightly short of the fully fine-tuned models, with an average drop of about 0.03 in safety score. For the 14B models, there is a consistent but small decrease in reasoning accuracy compared to the base model, while safety performance remains comparable to that of the fully fine-tuned model.

### 5 How Much LoRA is Enough?

In this section, we ablate the LoRA configuration to identify the key elements that matter most. We examine three factors: (1) the rank $r$, (2) the modules to which LoRA is applied, and (3) the layers to which LoRA is applied. Our goal is to determine the minimal setup needed for LoRA—i.e., the smallest update sufficient to achieve both strong reasoning and safety.

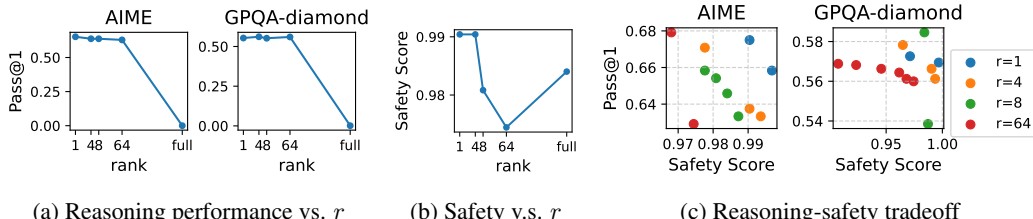

(a) Reasoning performance vs. $r$     (b) Safety v.s. $r$     (c) Reasoning-safety tradeoff

Figure 3: In (a) and (b), we show reasoning and safety performance at different LoRA ranks $r$ for the 14B model, respectively. Full-model fine-tuning is included as the rightmost point for reference. Reasoning performance decreases as $r$ increases, while safety first decreases and then increases. Overall, very low ranks are recommended, and $r = 1$ already achieves the best performance in both metrics. In (c), we visualize the Pareto frontiers of the reasoning–safety tradeoff when the training epoch is varied and observe that $r = 1$ is sufficient to achieve an excellent tradeoff, outperforming other ranks, especially on AIME.

To compare different LoRA configurations, we visualize performance in the reasoning–safety plane. For each configuration, we train for 10 epochs, which yields multiple checkpoints with varying reasoning and safety scores. We then evaluate configurations by examining the Pareto frontier of these checkpoints, which allows us to compare tradeoffs effectively.

## 5.1 RANK: $r = 1$ IS SUFFICIENT

We explore the effect of the rank $r$ in LoRA. Specifically, we run experiments with the 14B model by applying LoRA to the MLP layers while varying $r$. Results at the final checkpoint are shown in Figures 3a and 3b. We also include full-model fine-tuning as the rightmost point, since it allows for full-rank updates. Reasoning performance generally declines as $r$ increases, although the drop is minor between $r = 1$ and $r = 8$. For safety scores, we observe a notable decrease when $r$ increases from 4 to 64, whereas full-model fine-tuning yields a better safety score than $r = 64$. We suspect this may be due to optimization difficulties at intermediate high ranks, while very low-rank (under-parameterized) or full-rank (over-parameterized) setups may benefit from easier optimization dynamics, leading to better safety outcomes. Most importantly, we find that $r = 1$ is sufficient to achieve the best performance in both reasoning and safety. Figure 3c further confirms that $r = 1$ also achieves the best (on AIME) or nearly best (on GPQA) tradeoff when varying the number of training epochs. This indicates that we can achieve strong performance on both reasoning and safety at the lowest fine-tuning cost.

Additionally, the fact that $r = 1$ is sufficient reflects the inherently low-rank nature of the safety alignment task itself. This connects to prior work showing that safety can be mediated by a single direction within the model's internal representations, often referred to as a steering vector (Panickssery et al., 2023) or refusal features (Arditi et al., 2024; Yu et al., 2024). This perspective may explain why a rank-1 update is enough to achieve safety.

## 5.2 MODULES: UP-PROJECTION MATTERS MOST IN MLPS

The LoRA adapter is usually applied to attention layers and/or MLP modules, e.g., by setting `target_modules` when using the `PEFT` package. Here, we explore the effect of applying LoRA to different modules. We first compare applying it to both attention and MLP layers (QKVO & MLP in the figure) versus only applying it to MLP layers. Figure 4 shows the comparison for the 14B models with $r = 1$. We observe that applying LoRA only to the MLP layers yields a similar Pareto frontier compared to applying it to both modules.

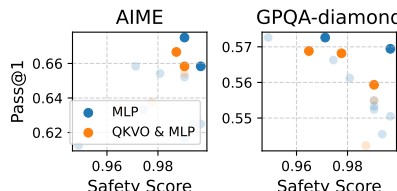

Figure 4: Applying LoRA to MLP modules alone is sufficient. Faded points indicate non–Pareto-frontier points.

Next, we perform an ablation over the modules within the MLP. Specifically, in the `Qwen` architecture, the MLP layers use the popular SwiGLU Chowdhery et al. (2022) structure that contains a gate

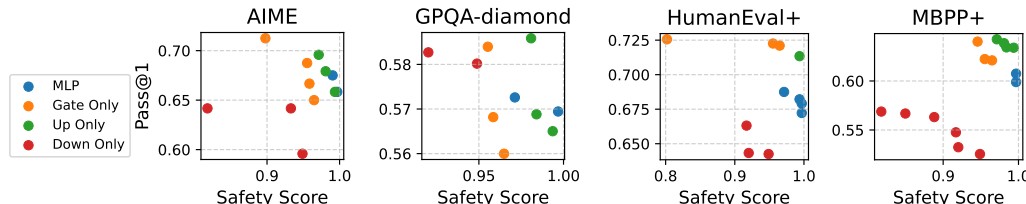

Figure 5: We compare applying LoRA to different projections within the MLP layers. The results show that applying it only to the up projection achieves the best tradeoff, and even outperforms applying it to the full MLP on the coding benchmarks.

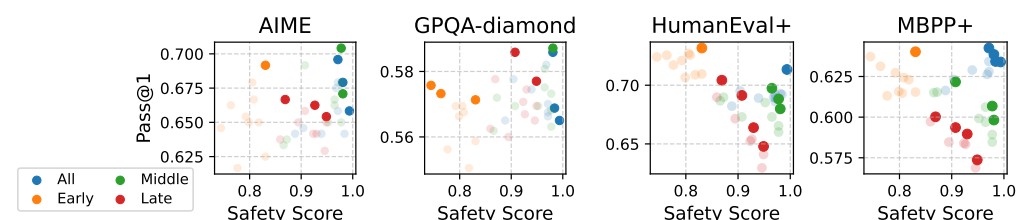

Figure 6: Applying LoRA to the middle layers (17–32) achieves a better tradeoff compared to using either the early or late layers, and performs on par with or only slightly behind using all layers across tasks. In the plot, the faded points indicate non–Pareto-frontier points.

projection, an up projection, and a down projection. We apply LoRA to only one of these projection layers at a time and evaluate the results. We use $r = 1$. Figure 5 shows that across different tasks, applying LoRA only to the up projection achieves strong results, with the Pareto frontier often on par with using the full MLP and even outperforming it on the two coding benchmarks. In contrast, applying LoRA only to the down projection yields noticeably worse performance. These findings suggest that different projections within the MLP contribute differently to the reasoning–safety tradeoff, and that the up projection is particularly important and sufficient by itself in our setup.

**Discussion.** We mainly focus our ablations on the MLP, as it is primarily responsible for feature transformations and thus well-suited for safety alignment. However, we believe it is also worthwhile to study ablations over the attention layers, which we leave for future work. It is also valuable to further investigate why the up projection alone is the best choice. Additionally, extending these experiments to other LLM architectures would be an important direction for future work.

### 5.3 LAYERS: MIDDLE LAYERS MATTER MOST

We ablate over layers in the model. The 14B `Qwen2.5` architecture has 48 layers in total, and we apply LoRA to only 16 of them. We consider three configurations: (1) layers 5–20, denoted as "Early Layers", (2) layers 17–32, denoted as "Middle Layers", and (3) layers 25–40, denoted as "Late Layers". In all cases, we apply LoRA only to the up projection layers with $r = 1$. The results are shown in Figure 6. Across tasks, we observe that applying LoRA to the middle layers yields the best tradeoff, achieving performance on par with using all layers on `AIME` and `GPQA`, and only slightly behind using all layers on `HumanEval+` and `MBPP+`. In contrast, applying LoRA to either the early or late layers results in a noticeably worse tradeoff. This shows that the middle layers are most important for balancing reasoning and safety. Interestingly, this again connects to prior findings on steering vectors (Panickssery et al., 2023) and refusal features (Arditi et al., 2024; Yu et al., 2024), which suggest that directions in intermediate representations responsible for safety behavior are most prominent in the middle layers.

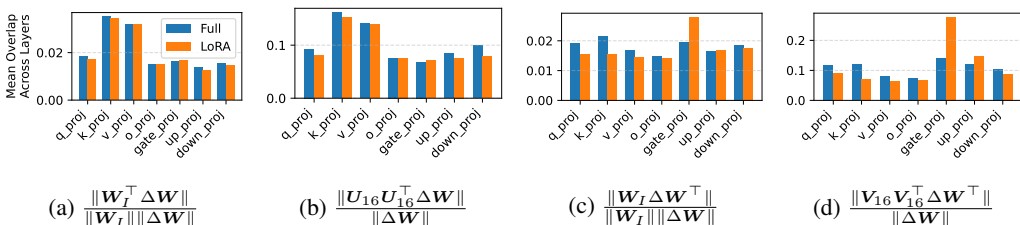

(a) $\frac{\|\boldsymbol{W}_I^\top \Delta \boldsymbol{W}\|}{\|\boldsymbol{W}_I\|\|\Delta \boldsymbol{W}\|}$    (b) $\frac{\|\boldsymbol{U}_{16}\boldsymbol{U}_{16}^\top \Delta \boldsymbol{W}\|}{\|\Delta \boldsymbol{W}\|}$    (c) $\frac{\|\boldsymbol{W}_I \Delta \boldsymbol{W}^\top\|}{\|\boldsymbol{W}_I\|\|\Delta \boldsymbol{W}\|}$    (d) $\frac{\|\boldsymbol{V}_{16}\boldsymbol{V}_{16}^\top \Delta \boldsymbol{W}^\top\|}{\|\Delta \boldsymbol{W}\|}$

Figure 7: LoRA updates exhibit smaller overlap with the original weights compared to the full-model fine-tuning updates. Although the reduction in overlap is sometimes small, it can be observed across most layers for all four metrics, which cover both the column (a)(b) and row spaces (c)(d). The 14B models are used here.

## 6 EXPLORING THE STRUCTURE OF LoRA WEIGHTS

### 6.1 LoRA UPDATES HAVE LESS ALIGNMENT WITH INITIAL WEIGHTS

Intuitively, if the initial (reasoning) weights $\boldsymbol{W}_I$ and the LoRA update $\Delta \boldsymbol{W}$ have only a small alignment, this suggests minimal interference between the safety-oriented update and the weights critical for reasoning. LoRA already constrains $\Delta \boldsymbol{W}$ to be low-rank, meaning it spans only a small subspace of the full weight space. We further examine the orientation of this subspace: how do the directions learned by LoRA compare to those spanned by $\boldsymbol{W}_I$? To quantify this, we compute the following four metrics: (1) $\frac{\|\boldsymbol{W}_I^\top \Delta \boldsymbol{W}\|}{\|\boldsymbol{W}_I\|\|\Delta \boldsymbol{W}\|}$, (2) $\frac{\|\boldsymbol{U}_{16}\boldsymbol{U}_{16}^\top \Delta \boldsymbol{W}\|}{\|\Delta \boldsymbol{W}\|}$, (3) $\frac{\|\boldsymbol{W}_I \Delta \boldsymbol{W}^\top\|}{\|\boldsymbol{W}_I\|\|\Delta \boldsymbol{W}\|}$, and (4) $\frac{\|\boldsymbol{V}_{16}\boldsymbol{V}_{16}^\top \Delta \boldsymbol{W}^\top\|}{\|\Delta \boldsymbol{W}\|}$. Here, $\boldsymbol{U}_{16}$ and $\boldsymbol{V}_{16}$ are matrices containing the top 16 left and right singular vectors of $\boldsymbol{W}_I$, respectively, obtained via truncated SVD. Intuitively, (1) and (2) capture the overlap between $\boldsymbol{W}_I$ and $\Delta \boldsymbol{W}$ in the column space: (1) is a matrix-level analogue of cosine similarity, while (2) measures the normalized projection of $\Delta \boldsymbol{W}$ onto the top dominant directions of $\boldsymbol{W}_I$. Similarly, (3) and (4) capture overlap in the row space. The column and row spaces correspond to the directions the matrices "write to" and "read from", respectively. A smaller value in any of these metrics indicates greater orthogonality between $\boldsymbol{W}_I$ and $\Delta \boldsymbol{W}$ in the corresponding space.

We compare the full-rank fine-tuned model with the LoRA fine-tuned model in which both the attention and MLP modules are updated with $r = 4$, making the two settings more comparable since updates occur in all major modules. We compute the alignment metrics for different module types across layers, average them over layers, and report the results for the 14B models in Figure 7. We observe that LoRA achieves smaller overlap in most modules across the metrics, with a few exceptions. This suggests that, for most weights, $\boldsymbol{W}_I$ and $\Delta \boldsymbol{W}$ are more orthogonal—both in the column and row spaces—for LoRA than for full-model fine-tuning. In other words, under LoRA fine-tuning, the safety-oriented updates read from and write into subspaces that are more separate from those used by the original reasoning-related weights—more so than in the full-model fine-tuned version. Although the reduction in alignment values is sometimes small, it may still indicate that LoRA updates interfere less with the reasoning-related components of the model, potentially explaining the better preservation of reasoning performance. A more in-depth investigation is needed to fully understand the underlying mechanisms and to develop more precise metrics for capturing this effect—an important direction for future work.

### 6.2 EXPLORING METHODS THAT FURTHER REDUCE ALIGNMENT

Given the observations in Section 6.1, we ask whether further reducing the overlap between $\Delta \boldsymbol{W}$ and $\boldsymbol{W}_I$ could lead to even better reasoning performance without compromising safety. This question is particularly relevant because, while the results achieved by LoRA in Section 4.4 are strong, they are not perfect—a small performance gap remains compared to the original reasoning model, especially for the 14B model on `AIME`, `HumanEval+`, and `MBPP+`.

We experiment with two approaches: (1) **Regularization during LoRA training:** adding a penalty to discourage overlap between $\boldsymbol{W}_I$ and $\Delta \boldsymbol{W}$, targeting either the column space (`reg-col`) or both the column and row spaces (`reg-both`). For efficiency, we approximate $\boldsymbol{W}_I$ via a truncated SVD. We tried different values of $\beta$ but observed negligible differences, so we fix $\beta = 1$.

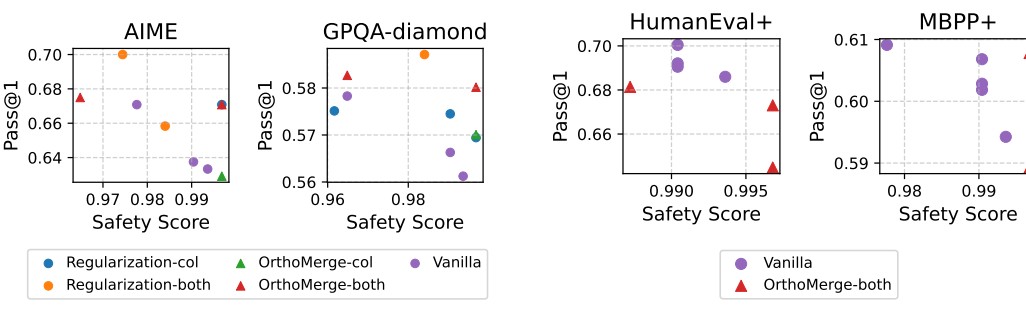

(a) 14B models on `AIME` and `GPQA`.          (b) 14B models on `HumanEval+` and `MBPP+`.

Figure 8: For each method that produces multiple checkpoints (e.g., across epochs or hyperparameter settings), we visualize the Pareto-frontier points. Methods enforcing orthogonality in both row and column spaces perform better than column-only variants. The post-hoc method `OrthoMerge-both` is the most promising, with points concentrated in the upper-right corner and its best point strictly dominating vanilla LoRA on `AIME` and `GPQA`. On `MBPP+`, it achieves a slightly better Pareto frontier, while on `HumanEval+` it slightly underperforms.

(2) **Post-hoc Orthogonalization:** post-processing $\Delta W$ before applying Equation 1, projecting it onto the orthogonal complement of the top-$k$ singular vectors of $W_I$. We test both column-only (`OrthoMerge-col`) and column+row (`OrthoMerge-both`) variants, with scaling $\lambda$ to offset safety loss. Further details are deferred to Appendix B.

Figure 8a shows the results on `AIME` and `GPQA` for the 14B models with $r = 4$. For each method that yields multiple checkpoints (e.g., different epochs for vanilla and regularization fine-tuning, or different hyperparameters for `OrthoMerge`), we visualize the Pareto-frontier points. We observe that methods addressing both the row and column spaces ("both") tend to yield better results than those that only operate on the column space ("col"). Among them, `OrthoMerge-both` appears most promising, with more points concentrated in the upper-right corner and its best point strictly dominating vanilla LoRA. Therefore, we additionally evaluate `OrthoMerge-both` on the coding benchmarks in Figure 8b, where it slightly underperforms on `HumanEval+` but slightly outperforms vanilla on `MBPP+` in terms of the tradeoff.

Overall, we observe modest yet inconsistent gains from post-hoc orthogonalization. This highlights the potential of controlling the subspace geometry of LoRA updates and points to the need for more nuanced methods that yield consistent improvements.

## 7 CONCLUSION

In this work, we identified a simple yet effective solution to the previously observed tension between reasoning and safety in LLMs. Through extensive experiments, we showed that applying LoRA during safety fine-tuning preserves reasoning capabilities while achieving strong safety alignment, in contrast to full-model fine-tuning, which significantly degrades reasoning. We further ablate the LoRA configuration to identify the key elements and make three important findings: (1) rank-1 updates are sufficient to achieve the best reasoning–safety tradeoff, (2) the up projection modules are most important for the tradeoff, yielding even better results than updating all or other modules, and (3) middle layers matter most, more than early or late layers. These results highlight that strong safety and reasoning can be achieved at minimal computational cost when updates are applied in the right places. Finally, we analyze the weight structure of LoRA and find that its updates are more orthogonal to the initial weights, which may help explain their reduced interference with reasoning. While our attempts to further enforce orthogonality yield only modest and inconsistent improvements, they highlight the potential of better controlling the geometry of safety updates as an avenue for future research. Overall, our results provide both a practical recipe for safety alignment of reasoning models and new insights into why LoRA is uniquely well-suited to this problem.

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

## A  EXPERIMENTAL DETAILS

### A.1  DETAILS ABOUT DIRECTREFUSAL

The original `DirectRefusal` dataset can be downloaded from `TianshengHuang/DirectRefusal` on Hugging Face. For each harmful request, the corresponding response includes a short thought—"I should not answer this question!"—followed by a refusal answer. We adapted the dataset to ensure it follows the `DeepSeek-R1` chat template.

### A.2  DETAILS ABOUT STRONGREJECT

For each question, we sample the answer using greedy decoding and pass the answer to `Llama-Guard-3-8B` to judge whether it is safe. Safety is determined by checking whether the substring "safe" or "unsafe" is contained in the evaluator's output.

### A.3  DETAILS ABOUT AIME AND GPQA

We adapted the codebase from Li et al. (2025b), which builds on top of `lm-evaluation-harness` Gao et al. (2024), to evaluate models on `GPQA` and `AIME`. For `AIME`, answers are evaluated using `Qwen2.5-32B-Instruct` as a judge. For `GPQA`, since it is a multiple-choice benchmark, answers are evaluated via regular expression matching. To improve evaluation accuracy, we made some small adjustments. One change is modifying the prompt to better ensure the model outputs answers in the desired format:

```
You are solving a multiple-choice question. At the end, present your
    final answer using the format: Final Answer: \boxed{X}, where X is
    one of A, B, C, or D.

Question: {{Question}}
Choices:
(A) {{choice1}}
(B) {{choice2}}
(C) {{choice3}}
(D) {{choice4}}
```

Another adjustment is in the answer extraction logic—we enhanced the original implementation to handle a wider range of answer formats that models produce.

During generation, we set the temperature to 0.6, top_p to 0.95, and the maximum number of generated tokens to 32,768.

### A.4  DETAILS ABOUT HUMANEVAL AND MBPP

We adapted the codebase of `EvalPlus` Liu et al. (2023). The original implementation included a response prefix, designed for earlier models that did not explicitly support an intermediate reasoning process. This prefix—such as "Below is a Python script with a self-contained function that efficiently solves the problem and passes corresponding tests:"—was prepended to model outputs during generation. However, this practice introduces unfair bias by encouraging all models to directly generate code. It inadvertently benefits the full-model fine-tuned baseline—which would otherwise often refuse to answer—by effectively forcing it to produce code. Conversely, it disadvantages reasoning-aligned models by disrupting the expected format that includes intermediate "thought", causing them to skip the thinking process entirely. This can result in skewed conclusions. We suspect this explains the abnormally low accuracy of the base reasoning model and the high accuracy of the full-model fine-tuned variant reported in Jiang et al. (2025). To address this issue, we remove the response prefix. We also slightly reword the prompt to better align with reasoning models.

During generation, we set the temperature to 0.6 and the maximum number of generated tokens to 32,768.

### A.5 TRAINING DETAILS

For 7B models, full-model fine-tuning is conducted using 4 GPUs with a batch size of 2 per device for 5 epochs. LoRA fine-tuning uses 2 GPUs with a batch size of 2 per device for 10 epochs. We set the LoRA hyperparameters as $\alpha = 16$ and `lora_dropout` = 0.05.

For 14B models, full-model fine-tuning is performed using 8 GPUs with a batch size of 1 per device for 5 epochs. LoRA fine-tuning uses 4 GPUs with a batch size of 2 per device for 10 epochs. We set the LoRA hyperparameters as $\alpha = 16$ and `lora_dropout` = 0.05.

Across all experiments, we use a learning rate of 5e-5 and a weight decay of 1e-4.

## B  DETAILED DESCRIPTION OF THE METHODS USED IN SECTION 6.2

We consider two approaches for enforcing stronger orthogonality: regularization during LoRA fine-tuning and enforcing orthogonality during merging LoRA weights, as we discuss next.

**Regularization during LoRA training.** We add a penalty term to the loss that discourages overlap between $\boldsymbol{W}_I$ [1] and $\Delta \boldsymbol{W}$, specifically:

- **Regularization-col:** $\beta(\frac{\|\boldsymbol{W}_I^\top \Delta \boldsymbol{W}\|}{\|\boldsymbol{W}_I\|\|\Delta \boldsymbol{W}\|})^2$ encourages orthogonality in the column space.

- **Regularization-both:** $\beta(\frac{\|\boldsymbol{W}_I^\top \Delta \boldsymbol{W}\|}{\|\boldsymbol{W}_I\|\|\Delta \boldsymbol{W}\|})^2 + \beta(\frac{\|\Delta \boldsymbol{W}^\top \boldsymbol{W}_I\|}{\|\boldsymbol{W}_I\|\|\Delta \boldsymbol{W}\|})^2$ encourages orthogonality in both column and row spaces.

We tried different values of $\beta$ but found no significant change in the results, so we fix $\beta = 1$.

**Enforcing Orthogonality During Merging LoRA Weights.** Starting with updates obtained from standard LoRA, we modify how they are merged with $\boldsymbol{W}_I$, which we call OrthoMerge. Before applying Equation 1, we preprocess $\Delta \boldsymbol{W}$ as follows:

- **OrthoMerge-col:** $\Delta \boldsymbol{W} \leftarrow (\boldsymbol{I} - \boldsymbol{U}_k \boldsymbol{U}_k^\top) \Delta \boldsymbol{W}$ enforces column-space orthogonality based on the rank-$k$ SVD of $\boldsymbol{W}_I$.
- **OrthoMerge-both:** $\Delta \boldsymbol{W} \leftarrow (\boldsymbol{I} - \boldsymbol{U}_k \boldsymbol{U}_k^\top) \Delta \boldsymbol{W} (\boldsymbol{I} - \boldsymbol{V}_k \boldsymbol{V}_k^\top)$ enforces orthogonality in both column and row spaces. We found that directly applying OrthoMerge-both leads to a drop in safety scores. To mitigate this, we further scale up the orthogonal complement with $\Delta \boldsymbol{W} \leftarrow \lambda(\boldsymbol{I} - \boldsymbol{U}_k \boldsymbol{U}_k^\top) \Delta \boldsymbol{W} (\boldsymbol{I} - \boldsymbol{V}_k \boldsymbol{V}_k^\top)$ to compensates for the loss in safety. We experiment with different values of $\lambda$ in the range $\{1, 1.15, 1.75, 1.2, 1.25\}$. We set $k = 64$.

For both approaches, we omit the row-space-only variant, as it showed no significant improvement in our experiments.

## C  ADDITIONAL FIGURES

Figure 9 shows the results on the base versions of `HumanEval` and `MBPP`.

## D  SUPPLEMENTARY RESULTS ADDED FOR REBUTTAL

### D.1  NEW MODEL ARCHITECTURE

We conducted new experiments with `DeepSeek-R1-Distill-Llama-8B`, which is based on the `Llama-3.1-8B architecture`. In Figure 10, we observe the same consistent pattern demonstrating the effectiveness of LoRA, showing that our conclusion generalizes to different architectures.

---

[1] To avoid out-of-memory issues during training caused by the large dimensionality of model weights, in implementation we use a low-rank approximation of $\boldsymbol{W}_I$ instead of the full matrix.

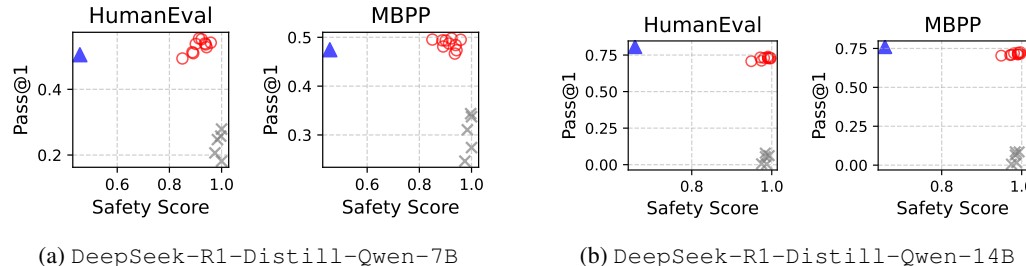

(a) `DeepSeek-R1-Distill-Qwen-7B`        (b) `DeepSeek-R1-Distill-Qwen-14B`

Figure 9: Results on the base versions of `HumanEval` and `MBPP` show the same pattern as in the plus versions shown in Figure 2, but with higher accuracy.

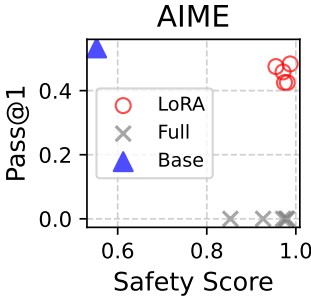

Figure 10: Safety vs. performance on `AIME` for `DeepSeek-R1-Distill-Llama-8B`. The results show the same pattern: LoRA fine-tuned models maintain strong performance in both safety and reasoning.

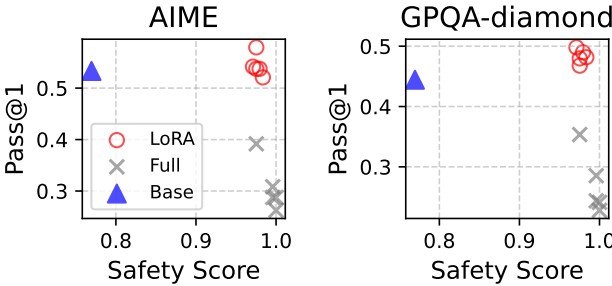

Figure 11: Results from evaluating safety on the `BeaverTails` dataset, showing that LoRA achieves strong safety across a broad evaluation set while preserving reasoning performance.

### D.2    NEW SAFETY EVALUATION DATASET

We conducted new experiments using the `BeaverTails` dataset, which covers 14 harm categories. In Figure 11, we observe the same consistent pattern: LoRA continues to achieve the best of both worlds. This further strengthens our conclusion and demonstrates the broad applicability of our findings across different safety evaluations.

### D.3    RESULTS ON NON-REASONING MODELS

We conducted new experiments on a non-reasoning, instruction-tuned model `Qwen2-1.5B-Instruct`. We performed safety fine-tuning, then measured utility using `BoolQ` and `COPA`, and measured safety using `StrongREJECT`. Interestingly, we observed a clear advantage for LoRA on `BoolQ`, but only marginal improvement on `COPA`. This suggests that the pattern may depend on the base model's capabilities, and future work can investigate why

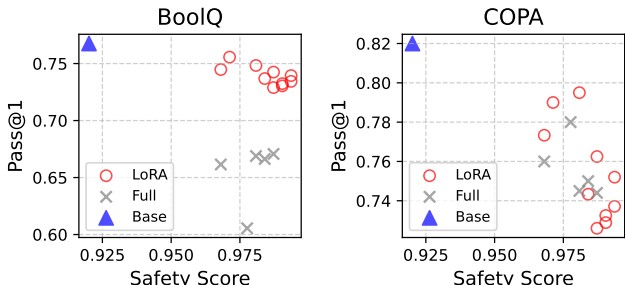

Figure 12: Results for the non-reasoning, instruction-tuned model `Qwen2-1.5B-Instruct`, with utility measured on `BoolQ` and `COPA`.

instruction-tuned models behave differently. Nevertheless, the contribution of this work—focused on reasoning models—remains significant.

