# OpenReview forum: "LoRA is All You Need for Safety Alignment of Reasoning LLMs"
_ICLR.cc/2026/Conference — Submitted to ICLR 2026_

### Official Review · Reviewer_DsR5 · 2025-10-31

**Soundness:** 2
**Presentation:** 3
**Contribution:** 2
**Rating:** 4
**Confidence:** 4

**Summary:**

This paper investigates whether the Safety Tax can be mitigated using only LoRA fine-tuning. Through extensive ablations on rank, module, and layer configurations, the authors show that low-rank LoRA updates can preserve reasoning performance while achieving strong safety alignment. They further analyze the geometric structure of LoRA updates, suggesting that their low alignment with reasoning weights explains the reduced interference and improved reasoning–safety balance.

**Strengths:**

- Demonstrates that LoRA-only fine-tuning can mitigate the Safety Tax while preserving reasoning ability — an interesting and practical finding.
- Well-designed ablations reveal key factors (rank = 1, MLP up-projection, middle layers) that contribute most to the reasoning–safety trade-off.
- Provides a geometric perspective on why LoRA interferes less with reasoning, through alignment and subspace analyses.

**Weaknesses:**

- The paper makes a strong claim that “LoRA is all you need” to address the safety–reasoning trade-off. While the presented results are intriguing, the current experimental scope is insufficient to substantiate this claim. A wider range of backbones and model sizes should be evaluated to demonstrate consistency across architectures and scales (R1-1.5 ~ R1-32B, s1, Qwen3, etc).

- Moreover, prior analyses (Jain et al., 2024; Wei et al., 2024) about low-rank safety directions generalize to general LLMs, not just reasoning-centric LRMs. Therefore, the authors’ argument should extend beyond LRMs to general LLMs. Since the Safety Tax is not unique to reasoning models but also arises in other capabilities, the proposed hypothesis should be validated across diverse tasks beyond reasoning.

- In Figure 7, the alignment difference between full fine-tuning and LoRA appears marginal, which weakens the causal claim.

- The trade-off plots contain multiple points per configuration, yet their meanings are under-explained (e.g., epoch checkpoints, random seeds, or hyperparameter variations). The large variance across these points also raises concerns about training stability and reproducibility.

- Finally, the evaluation would be more convincing if it included adaptive or adversarial safety tests—such as obfuscation, paraphrased jailbreaks, or multi-turn exploit prompts—that better reflect real-world attack surfaces and robustness challenges.

[1] Jain, Samyak, et al. "What makes and breaks safety fine-tuning? a mechanistic study." Advances in Neural Information Processing Systems 37 (2024): 93406-93478.

[2] Wei, Boyi, et al. "Assessing the brittleness of safety alignment via pruning and low-rank modifications." arXiv preprint arXiv:2402.05162 (2024).

**Questions:**

See Weaknesses

---

> ### Author Response · Authors · 2025-11-26
>
> > Other family of models and architectures
>
> The models we use in our paper are R1-Distill-7B, which is based on Qwen2.5-Math-7B, and R1-Distill-14B, which is based on Qwen2.5-14B. We note that most existing open-weight reasoning models (e.g., S1, Qwen 3, Bespoke-Stratos) share the same underlying architecture—namely, the Qwen architecture—we expect the observations we made to generalize broadly. **Additionally, we conducted new experiments** with DeepSeek-R1-Distill-Llama-8B, which is based on the Llama-3.1-8B architecture. The results are presented in **Figure 10 in Appendix D of the revised PDF**. We observe the same consistent pattern demonstrating the effectiveness of LoRA, showing that our conclusion generalizes to different architectures.
>
> > Additional benchmarks for safety evaluation
>
> **We conducted new experiments using the BeaverTails dataset**, which covers 14 harm categories. The results are presented in **Figure 11 in Appendix D of the revised PDF**. We observe the same consistent pattern: LoRA continues to achieve the best of both worlds. This further strengthens our conclusion and demonstrates the broad applicability of our findings across different safety evaluations.
>
>
>
> > Moreover, prior analyses (Jain et al., 2024; Wei et al., 2024) about low-rank safety directions generalize to general LLMs, not just reasoning-centric LRMs. Therefore, the authors’ argument should extend beyond LRMs to general LLMs. Since the Safety Tax is not unique to reasoning models but also arises in other capabilities, the proposed hypothesis should be validated across diverse tasks beyond reasoning.
>
> First, our work specifically focuses on reasoning models, as they are both emergent and important in current LLM development. Whether the same conclusions hold for other types of models or tasks is beyond the scope of this work and does not weaken our contribution. Importantly, as discussed in our related work section, there is a substantial distinction between reasoning fine-tuning and instruction fine-tuning: reasoning training fundamentally reshapes the model’s internal representations, whereas instruction tuning typically introduces only small shifts. Their safety behavior can therefore differ significantly. That said, exploring this question for non-reasoning models is still interesting. As a preliminary exploration, we conducted new experiments on a non-reasoning, instruction-tuned model (Qwen2-1.5B-Instruct). We performed safety fine-tuning, then measured utility using BoolQ and COPA, and measured safety using StrongREJECT. The results are presented in Figure 12 in Appendix D of the revised PDF. Interestingly, we observed a clear advantage for LoRA on BoolQ, but only marginal improvement on COPA. This further demonstrates that the pattern may differ between non-reasoning and reasoning models.
>
> > In Figure 7, the alignment difference between full fine-tuning and LoRA appears marginal, which weakens the causal claim.
>
> While the alignment differences in Figure 7 are small in magnitude, they are consistent across modules, indicating a structural difference between LoRA and full-model fine-tuning updates. This provides evidence that LoRA modifies the model in a more targeted and less interfering manner, which aligns with our broader finding that LoRA preserves reasoning ability while full fine-tuning disrupts it. The purpose of this section is to highlight this structural distinction, providing interpretive insight into the effectiveness of LoRA.
>
> > The trade-off plots contain multiple points per configuration, yet their meanings are under-explained (e.g., epoch checkpoints, random seeds, or hyperparameter variations). The large variance across these points also raises concerns about training stability and reproducibility.
>
> The explanation of these points is already clearly stated in the paper. For Figure 2, as noted in the caption (line 236), the points correspond to different epochs, and the consistent advantage of LoRA across epochs is clearly visible. In Section 5, because we compare different LoRA configurations that may yield similar results, we present the Pareto frontier for each configuration to enable a more informative comparison. As explained in lines 286–290, Figures 3–6 visualize only the Pareto-frontier checkpoints (across epochs) for each setup. This allows us to see which configuration achieves the best tradeoff when the number of training epochs varies.
>
> We hope the reviewer finds our clarifications satisfactory and will update the score accordingly.

---

### Official Review · Reviewer_twiY · 2025-11-01

**Soundness:** 2
**Presentation:** 3
**Contribution:** 2
**Rating:** 4
**Confidence:** 3

**Summary:**

This  paper explores how Lora performs on safety alignment task and find that it can effective improve safety score without droping reasoning ability.

**Strengths:**

(1) This empirical study is careful and fairly comprehensive. Results on multiple benchmarks are reported.

(2) The findings that Lora can successfully avoid the trade-off between reasoning ability and model safety is interesting and useful.

(3) This paper is well-structured and easy to follow.

**Weaknesses:**

(1) The benchmarked models lack diversity — all reasoning models are derived from DeepSeek. It remains unclear whether the findings generalize to other reasoning models or architectures, such as GPT-OSS-20B or GPT-OSS-120B.

(2) This paper lacks theoretical analysis explaining why the LoRA technique can mitigate the “safety tax” issue. A deeper investigation or theoretical justification would strengthen the claims.

(3) The safety evaluation pipeline may have limitations. Safety is automatically assessed using Llama-Guard-3-8B, which could introduce bias. Incorporating multi-metric safety evaluations (e.g., jailbreak or red-teaming tests) would provide more comprehensive and convincing evidence.

**Questions:**

(1) Can the main findings generalize to other reasoning models such as GPT-OSS-20B?

(2) In Figure 3(b), why does r = 64 yield the lowest model safety performance? Is this a consistent phenomenon observed across different models? The paper explains that the “middle-rank” setting might be harder to optimize, but is there any empirical or theoretical evidence supporting this explanation?

---

> ### Author Response · Authors · 2025-11-26
>
> > Generalisability of the findings to other models
>
> (1) Since most existing open-weight reasoning models (e.g., S1, Qwen 3, Bespoke-Stratos) share the same underlying architecture—namely, the Qwen architecture—we expect the observations we made to generalize broadly. (2) In our environment, the pre-release vLLM wheel for GPT-OSS could not be installed due to an unavailable pinned PyTorch nightly; setting up a custom source build / Docker stack was out of scope for the rebuttal timeline. (3) **We conducted new experiments** with DeepSeek-R1-Distill-Llama-8B, which is based on the Llama-3.1-8B architecture. The results are presented in **Figure 10 in Appendix D of the revised PDF**. We observe the same consistent pattern demonstrating the effectiveness of LoRA, showing that our conclusion generalizes to different architectures.
>
> > “… Safety is automatically assessed using Llama-Guard-3-8B, which could introduce bias … multi-metric safety evaluations …”
>
> (1) Regarding the choice of Llama-Guard-3-8B, as mentioned in our paper, it is already the strongest available evaluation strategy compared to other methods, according to the study in Jiang et al. (2025). (2) Additionally, **we conducted new experiments using the BeaverTails dataset**, which covers 14 harm categories. The results are presented in **Figure 11 in Appendix D of the revised PDF**. We observe the same consistent pattern: LoRA continues to achieve the best of both worlds. This further strengthens our conclusion and demonstrates the broad applicability of our findings across different safety evaluations.
>
> > This paper lacks theoretical analysis explaining why the LoRA technique can mitigate the “safety tax” issue. A deeper investigation or theoretical justification would strengthen the claims.
>
> Empirical investigation is a standard and valuable approach in this area. To the best of our knowledge, there is currently no theoretical framework that explains the safety–reasoning tradeoff in reasoning models. Thus, empirical analysis is an appropriate methodology at this stage and does not weaken our contributions. The effectiveness of LoRA—together with the additional insights from our targeted ablations—remains both novel and significant. One key intuition supported by our results is that safety-related updates and reasoning-related weights lie in largely orthogonal directions, which explains why low-rank updates can improve safety without harming reasoning. Additionally, Sec 6 provides preliminary empirical evidence consistent with this intuition—showing, for example, reduced alignment between LoRA updates and the initial weights. These observations offer meaningful interpretive insight and complement our empirical findings.
>
> > Question about middle rank setting yielding lower safety performance
>
> We will add the following reference to our discussion in lines 299–300 regarding why r=64 yields lower safety performance. It is known in the literature that heavy over-parameterization (e.g., neural networks whose width is polynomial in the sample size) can make the optimization landscape more favorable, enabling SGD to efficiently converge to global minima [1]. Many theoretical results on optimization guarantees also rely on over-parameterization assumptions [2,3,4]. When the rank is extremely small, the optimization problem becomes highly under-parameterized and therefore structurally simple. The “middle-rank’’ regime may lack both the simplicity of low-rank settings and the optimization benefits of strong over-parameterization, making it potentially harder to optimize.
>
> [1] Allen-Zhu, Zeyuan, Yuanzhi Li, and Zhao Song. "A convergence theory for deep learning via over-parameterization." International conference on machine learning. PMLR, 2019.
>
> [2] Zou, Difan, and Quanquan Gu. "An improved analysis of training over-parameterized deep neural networks." Advances in neural information processing systems 32 (2019).
>
> [3] Arora, Sanjeev, et al. "Fine-grained analysis of optimization and generalization for overparameterized two-layer neural networks." International conference on machine learning. PMLR, 2019.
>
> [4] Du, Simon, et al. "Gradient descent finds global minima of deep neural networks." International conference on machine learning. PMLR, 2019.
>
> We hope the reviewer finds our clarifications satisfactory and will update the score accordingly.

---

### Official Review · Reviewer_4WFt · 2025-11-03

**Soundness:** 2
**Presentation:** 3
**Contribution:** 2
**Rating:** 2
**Confidence:** 4

**Summary:**

The paper targets "Safety Tax", which is preserving the model's reasoning capacity while aligning for safety. The paper utilizes LoRA on a refusal dataset to effectively align the model for safety while preserving the reasoning capacity. The authors show that using r=1 rank achieves the best reasoning-safety tradeoff.

**Strengths:**

- The paper shows simple yet effective usage of LoRA in finetuning for safety alignment.
- The results show that the LoRA-trained model is both safe and has high reasoning performance in different benchmarks.

**Weaknesses:**

- The paper does not exhibit weight-level selectivity; instead, it adopts a more coarse-grained perspective, assuming that all parameters contribute collectively to the model’s reasoning capability. The selectivity applied is primarily at the layer or module level.

- The paper lacks of theoretical ground for its claims and has an experimental approach.

- The proposed post-hoc method doesn't improve the reasoning performance, but the authors also claim that the method needs more development.

- To me, the paper needs a more professional tone. The narration should support the claims with references or experimental data, which are given in lines 184-187.

- There is only one dataset used for the safety evaluation.

- The paper focuses on only one family of models.

**Questions:**

- Are the findings true for the different architectures?
- Why is only one family of models selected?
- Can authors compare their approach with other fine-tuning safety-alignment methods?

---

> ### Author Response · Authors · 2025-11-26
>
> > Additional safety evaluation dataset
>
> **We conducted new experiments using the BeaverTails dataset**, which covers 14 harm categories. The results are presented in **Fig 11 in Appendix D of the revised PDF**. We observe the same consistent pattern: LoRA continues to achieve the best of both worlds. This further strengthens our conclusion and demonstrates the broad applicability of our findings across different safety evaluations.
>
> > Other family of models and architectures
>
> The models we use in our paper are R1-Distill-7B, which is based on Qwen2.5-Math-7B, and R1-Distill-14B, which is based on Qwen2.5-14B. We note that most existing open-weight reasoning models (e.g., S1, Qwen 3, Bespoke-Stratos) share the same underlying architecture—namely, the Qwen architecture. This is why we expect our conclusions to generalize broadly to these models as well. **Additionally, we conducted new experiments** with DeepSeek-R1-Distill-Llama-8B, which is based on the Llama-3.1-8B architecture. The results are presented in **Figure 10 in Appendix D of the revised PDF**. We observe the same consistent pattern demonstrating the effectiveness of LoRA, showing that our conclusion generalizes to different architectures.
>
> > “The paper does not exhibit weight-level selectivity …”
>
> We believe the reviewer may have misunderstood the intention of our work. Our goal is not to identify individual weight-level contributions, which belongs to the interpretability literature and lies outside the scope of LoRA-based fine-tuning. Rather, our goal is to demonstrate the strong effectiveness of LoRA in balancing safety and reasoning—an outcome that is itself a significant and novel contribution. In addition, Sections 5.2 and 5.3 explicitly compare applying LoRA to different modules and layers, revealing clear selective structure: (1) the up-projection in MLPs is the most important component, and (2) the middle layers matter most. This finding is both novel and practically impactful, as it shows that the safety–reasoning tradeoff can be resolved with very small and targeted updates. Identifying individual weight-level contributions, while interesting, is beyond the scope of this work and not directly compatible with LoRA, which operates through low-rank updates rather than weight-level sparsity.
>
>
> > “The paper lacks of theoretical ground for its claims and has an experimental approach”
>
> Empirical investigation is a standard and valuable approach in this area. To the best of our knowledge, there is currently no theoretical framework that explains the safety–reasoning tradeoff in reasoning models. Thus, empirical analysis is an appropriate methodology at this stage and does not weaken our contributions. The effectiveness of LoRA—together with the additional insights from our targeted ablations—remains both novel and significant. One key intuition supported by our results is that safety-related updates and reasoning-related weights lie in largely orthogonal directions, which explains why low-rank updates can improve safety without harming reasoning. Additionally, Section 6 provides preliminary empirical evidence consistent with this intuition—showing, for example, reduced alignment between LoRA updates and the initial weights. These observations offer meaningful interpretive insight and complement our empirical findings.
>
> > The proposed post-hoc method doesn't improve the reasoning performance, but the authors also claim that the method needs more development.
>
> We believe there is a misunderstanding of our goal and contributions. Our goal was not to claim a fully solved method, but to show that controlling alignment can matter for the safety–reasoning tradeoff. Our results demonstrate that this direction is promising and highlights where further development may be fruitful. We also believe it is valuable to report both what worked and what did not. In particular, we compared multiple regularization-based methods with post-hoc approaches and found that the post-hoc method can yield additional gains on AIME and GPQA. These results can meaningfully inform future research seeking stronger or more reliable approaches. We hope this clarifies the contribution of this exploratory analysis.
>
> > To me, the paper needs a more professional tone …
>
> We will revise the tone in the sections the reviewer suggested, making the statements more rigorous. While we have intuition that similar conclusions may hold for RL for the same underlying reasons, we agree that exact experiments are needed, and we consider this an interesting direction for future work.

---

> > ### Author Response · Authors · 2025-11-26
> >
> > > Can authors compare their approach with other fine-tuning safety-alignment methods?
> >
> > (1) To our knowledge, there are no existing fine-tuning–based safety-alignment methods that specifically focus on the tension between reasoning ability and safety for reasoning models. We refer the reviewer to the related work section for more discussion. (2) As shown in our experiments (Fig 2), simply applying LoRA already approaches the empirical ceiling in both safety and reasoning: safety that matches the full-model fine-tuned model and reasoning that matches the base reasoning model.
> >
> > We appreciate the reviewer’s feedback and hope the clarifications provided here justify an improved score.

---

### Meta-Review · Area_Chair_Cd7q · 2026-01-02

**Summary:**

This paper applies LoRA fine-tuning to a refusal dataset with the goal of balancing safety and preserving reasoning capability.

The reviewers raised several concerns:
(1) Multiple reviewers noted that the experimental evaluation is limited. The set of evaluated models lacks diversity, some claims are not fully supported by experimental evidence, and some reported improvements appear marginal.
(2) Several reviewers pointed out that the paper lacks sufficient theoretical justification.
(3) Evaluating safety solely using Llama-Guard-3-8B may introduce bias. Reviewers suggested incorporating more diverse evaluation methods, such as adaptive or adversarial safety testing.
(4) Restricting the analysis to reasoning-focused models was viewed as overly narrow.
(5) The LoRA updates were considered too coarse-grained, as they lack weight-level selectivity.

**Reviewer Concerns:**

The authors conducted additional experiments on other models and introduced new approaches for assessing safety; as a result, concerns (1) and (3) are partially addressed. However, after reading the revised paper, I concur with the reviewers that a more thorough empirical analysis would strengthen the work. While I agree with the authors that a formal theoretical framework is not strictly necessary, theory can often be helpful in clarifying and supporting key claims. In the absence of a theory, it would be particularly valuable to further substantiate claims such as "safety-related updates and reasoning-related weights lie in largely orthogonal directions."

**Reviewer Scores:**

I would say that the reviewers are unlikely to significantly increase their scores. While some concerns have been partially addressed, the additional experimental validation provided during the rebuttal does not appear to substantially change the paper.

---

### Decision · Program_Chairs · 2026-01-26

Reject